# Physiologically Based Pharmacokinetic Modeling of Biologic Case Studies in Monkeys and Humans Reveals the Necessity of an Additional Clearance Term

**DOI:** 10.3390/pharmaceutics17050560

**Published:** 2025-04-24

**Authors:** Felix Stader, Pradeep Sharma, Weize Huang, Mary P. Choules, Marie-Emilie Willemin, Xinwen Zhang, Estelle Yau, Abdallah Derbalah, Adriana Zyla, Cong Liu, Armin Sepp

**Affiliations:** 1Certara Predictive Technologies, Certara UK Ltd., Sheffield S1 2BJ, UKcong.liu@certara.com (C.L.); armin.sepp@certara.com (A.S.); 2Clinical Pharmacology and Quantitative Pharmacology, Biopharmaceuticals R&D, AstraZeneca, Cambridge CB1 9JS, UK; 3Genentech Inc., South San Francisco, CA 94404, USA; 4Astellas Pharma Global Inc., Northbrook, IL 60062, USA; 5Johnson & Johnson, 2340 Beerse, Belgium; 6Amgen Inc., South San Francisco, CA 94080, USA; 7Sanofi R&D, 94403 Vitry-sur-Seine, France

**Keywords:** physiologically based pharmacokinetic model, monoclonal antibody, antibody–drug conjugate, bispecific antibody, translational study

## Abstract

**Background/Objectives**: Physiologically based pharmacokinetic (PBPK) modeling is an important tool in biologic drug development. However, a standardized modeling strategy is currently missing. A cross-industry collaboration developed PBPK models for seven case studies, including monoclonal antibodies, antibody–drug conjugates, and bispecific T-cell engagers, to identify key parameters and establish a workflow to simulate biologic drugs in monkeys and in humans. **Methods**: PBPK models were developed in the monkey with limited data, including the molecular weight, the binding affinity to FcRn, and the additional systemic clearance of IgG, which is 20% of the total clearance. The binding affinity was only available for human FcRn and corrected for the known species-dependent differences in IgG binding. The strategy of monkey simulations was evaluated with an additional 14 studies published in the literature. Three different scenarios were simulated in humans afterwards: without, with allometrically scaled, and with optimized additional systemic clearance. **Results**: The plasma peak concentration and the area under the curve were predicted within 50% of the observed data for all studied case examples in the monkey, which demonstrates that sparse input parameters are sufficient for successful predictions in the monkey. Simulations in humans demonstrated the need for additional systemic clearance, because drug exposure was highly overpredicted without an additional systemic clearance term. Allometric scaling improved the predictions, but optimization led to the best fit, which is currently a limitation in the translation from animals to humans. **Conclusions**: This work highlights the importance of understanding the general mechanisms of drug uptake in different tissue types and cells in both target-dependent and -independent processes.

## 1. Introduction

Physiologically based pharmacokinetic (PBPK) modeling is an important tool in the development of therapeutic proteins (TPs) and oligonucleotides. It can be used throughout the cycle of drug development, starting from the identification of a “druggable” target, which requires a sufficient drug concentration under physiological conditions and the affinity of the drug to bind to the target to have a desired pharmacological effect. The in vitro measured drug characteristics, such as the target binding affinity, can be entered into a PBPK model to predict the result of any engineered modifications of the drug under physiological conditions. Thus, the PBPK approach can support drug design from a physiological perspective. Furthermore, potential off-site effects due to target expression at multiple locations in the body can be investigated by a PBPK model [1]. Later in drug development, the PBPK approach can be applied to predict the bioavailability after subcutaneous (SC) administration [2] or the dose in patient populations with limited clinical data, such as children [3]. Additionally, the drug–drug interaction likelihood between the released payload of an antibody–drug conjugate (ADC) and a co-medication can be analyzed by a PBPK model [4].

Typically, information about biologic drugs is first available in animals, leading to the development of preclinical PBPK models. Processes that determine the disposition of biologic drugs, such as the two-pore hypothesis [5] and the neonatal Fc receptor (FcRn) salvage pathway [6], are assumed to be highly conserved across species. Therefore, the same PBPK model structure, informed by different physiological databases for distinct animal species and humans, can be used for preclinical and human simulations [7]. However, the binding of immunoglobulin G (IgG) to FcRn is species-dependent [8]. During the development of novel biologic drugs, the binding is often only measured for human FcRn for capacity and cost reasons, and appropriate data are lacking in animals.

The PBPK models can be constantly revised and expanded during the drug development cycle when more experimental data become available. The goal is to develop a mechanistic model that achieves precise predictions that can eventually be used to predict the pharmacokinetics in humans. Currently, a standardized PBPK modeling strategy is missing. A cross-industry collaboration was formed to develop PBPK models for different biological moieties, including monoclonal antibodies (mAbs), ADCs, and bispecific T-cell engagers (Bi-TCEs). The objectives of the study were to (1) establish a workflow to develop a PBPK model for biologic drugs, (2) to analyze whether PBPK models for cynomolgus monkeys can be developed with sparse model parameterization, and (3) to investigate the need for additional systemic clearance in humans.

## 2. Materials and Methods

A cross-industry collaboration was established to seek case studies of biologic drugs from different companies. The inclusion criteria for case studies were a TP (1) independent of its size, (2) elimination pathways, (3) target-mediated drug disposition (TMDD) effects, or (4) route of administration. Preclinical data in cynomolgus monkeys and/or clinical data were required for performance evaluation. At least one dose had to be high enough to saturate the target to allow the optimization of the additional systemic clearance.

A total of seven case studies were collected, including five for mAbs, one for ADCs, and one for Bi-TCEs (Table 1 and Appendix A). The mAbs and the ADC were administered by intravenous infusion (IV) and the Bi-TCEs were administered SC. Human pharmacokinetic data were available for all seven case studies, but data in cynomolgus monkeys were only available for three mAbs. Therefore, a literature search was conducted to find more case examples of humanized IgG-like antibodies, administered IV, without target binding or with sufficiently high doses to saturate the target in the monkey. The keywords used were “cynomolgus monkeys” and “humanized IgG”. An additional 14 studies on the pharmacokinetics of humanized IgG in the monkey were found in the literature and used in the analysis (Appendix A).

Simulations were performed in the Simcyp Simulator^®^ V21 (Certara Predictive Technologies, Certara UK, Sheffield, UK), using either the Monkey Simulator for preclinical data or the Human Simulator for clinical data. The full PBPK distribution model was used in all cases, representing 15 different compartments, namely the lung, adipose tissue, bone, brain, heart, kidneys, muscle, skin, gut, pancreas, spleen, liver, a central lymph node compartment, and the venous and the arterial blood pool [9]. The model structure for TPs is permeability-limited, meaning that tissue is separated into the vascular space representing the blood vessels, the endosome of the endothelial cell layer, the interstitial space, and the intracellular space [9,10].

Distribution processes include the two-pore hypothesis to calculate the paracellular distribution between the vascular and the interstitial space [5]. The parameters for the two-pore hypothesis, including the determination of pore sizes and the number of pores, were described previously [9]. The transcellular distribution pathway consists of a linear rate, representing macropinocytosis into the endothelial cell layer, binding to FcRn at pH 6.0, and the recycling of the bound complex to the vascular space or transcytosis to the interstitial space. The pinocytosis uptake rate was assumed to be the same for all tissue types and was set to 0.0298 1/h based on published data on horseradish peroxidase, a common pinocytosis marker [11]. The tissue-specific FcRn abundance for humans was taken from the literature [12], and the same values per gram tissue were assumed to apply in the monkey in the absence of relevant data. The tissue-specific FcRn abundance in the monkey was used to calculate the total body FcRn concentration, which was used in the monkey simulations. The recycling rate was fitted to match the plasma concentration and half-life of exogenous IgG in monkeys [13] and humans [14].

Three unspecific elimination pathways were considered. The first was catabolism in endothelial cells of the unbound IgG-like antibody [10]. The contribution of each tissue to the catabolism was taken from the literature [15,16]. The second was additional plasma clearance, which represents processes that are not mechanistically accounted for, such as catabolism in antigen-presenting cells (APCs) or pinocytosis into non-FcRn expressing cells with subsequent catabolism. The additional systemic clearance of IgG was set to 20% of the total IgG clearance, based on previous findings [10]. Monkey simulations used the default value for the additional systemic clearance of IgG to investigate the minimal parameter requirement. The third case was renal filtration, which was only used for the Bi-TCE model [17].

The proposed workflow was based on the “learn and confirm” approach (Figure 1). Each company in the cross-industry collaboration developed their own model based on in vitro data. The required input parameters were the molecular weight of the TP and the binding affinity to FcRn at pH 6.0 (Table 2). Ideally, the measurement would be required for monkey FcRn and human FcRn, but, in all cases, measurements were only available for human FcRn. The species-dependent difference in IgG binding [8] was assumed to hold true in all cases, meaning that measured human KD values were multiplied by 0.45 to describe the binding to monkey FcRn. PK profiles in monkeys and in humans were used to evaluate the bottom-up predictions from in vitro data. If the predictions do not match the observed data, the learn and confirm approach can be used to build a more robust mechanistic model by performing additional in vitro measurements. Alternatively, processes that are not well understood, such as the additional systemic clearance, can be optimized. In this study, additional in vitro data were not obtained, but the additional clearance was investigated in the human simulations. Target binding can be included if the doses are not high enough to saturate the target or the animal species is cross-reactive.

The primary aim of this work was to evaluate the proposed simulation strategy and required input parameters to perform PBPK modeling for TPs. Simulations were first conducted in a typical male cynomolgus monkey, weighting 4 kg. Parameterization of the biologics drug in the monkey model was performed with minimal input data, including the molecular weight, the binding affinity to FcRn, and the additional systemic clearance of IgG. The monkey was a non-cross-reactive species for all investigated antibodies; thus, TMDD was not considered in the simulations.

In a second step, the model was translated to humans by running three different scenarios: no additional plasma clearance (default simulation), additional plasma clearance allometrically scaled from cynomolgus monkeys, and fitted plasma clearance (optimized simulation). To optimize the additional systemic clearance, the dose had to be high enough to saturate the target; thus, TMDD was not the clearance-determining process. Simulations in humans were performed in a virtual population of healthy volunteers with matching demographics compared with the clinical study that was mimicked regarding the number of subjects in the virtual trial, the proportion of females, and the age range. The dosing regimen (IV or SC administration; Appendix A) in the virtual trial was driven by the observed clinical study.

The prediction error (*PE*) was calculated for the peak concentration (C_max_) and the area under the curve to the last time point (AUC_last_) following$$PE(\%)=\left(\frac{predicted}{observed}\cdot 1\right)\cdot 100$$

A *PE* within the −50% to +50% margin was considered acceptable.

## 3. Results

Firstly, simulations were performed in cynomolgus monkeys with sparse data, including the molecular weight, the binding affinity to human FcRn scaled to monkey FcRn, and the additional systemic clearance of IgG (Figure 2). The mean *PE* for C_max_ across the different biologic drugs was 12.1% (min–max: −20.2–+64.1%), demonstrating a slight tendency towards overprediction. The C_max_ of one IgG-like antibody was outside the 50% margin but still within the 100% margin. The mean *PE* for the AUC across all studies was −1.9% (min–max: −37.0–+39.4%), demonstrating no systematic over- or underprediction of the drug exposure in cynomolgus monkeys, despite the minimal data that were used for the parameterization of the monkey model.

After the successful simulation in cynomolgus monkeys, predictions were performed in humans (Figure 3 and Figure 4). The mean PEs for the C_max_ of the default and optimized runs were 15.4% (min–max: −7.8–+51.0%) and 12.7% (min–max: −7.9–+54.0%), respectively. Similarly to the monkey, there was a tendency to overpredict the C_max_; however, apart from that of one IgG-like antibody, there was no *PE* above 50%. The AUC of the default predictions was 253.9% (min–max: +53.2–+617.0%), which showed a clear tendency for overprediction. Optimized additional systemic clearance therefore appears to be key for a successful prediction. After the optimization, the mean *PE* for the AUC was 2.8% (min–max: −17.0–+44.3%). Notably, drug exposure was predicted within the 50% margin of the observed data for the ADC and the Bi-TCEs. The additional systemic clearance could be scaled allometrically if a model can be developed in animals first. We had three mAbs with data in both monkeys and humans (Table 1). The *PE* of the AUC, simulated in humans, for the default case was 103.6%, 122.0%, and 347.9%; for the allometrically scaled clearance, it was 44.5%, 12.87%, and 36.7%; and, for the optimized clearance, it was −4.7%, 8.0%, and −9.8%, respectively.

## 4. Discussion

A cross-industry collaboration was established to utilize a workflow for PBPK model development for different TPs, including mAbs, ADCs, and Bi-TCEs. The study demonstrated that minimal input parameters (molecular weight, binding affinity to FcRn, additional systemic clearance of IgG) are required to achieve simulations in the monkey within 50% of the observed data. The binding affinity of a mAb to FcRn can be measured in humans only, and the scaling factor to monkey FcRn of 0.45 [8], multiplied by the experimental binding affinity to human FcRn, leads to successful predictions in the monkey.

All simulations, independently of the simulated TP, required an additional clearance pathway for monkeys and humans, beyond lysosomal catabolism and renal filtration. Simulations of Bi-TCEs revealed that predictions without the additional systemic clearance fell mostly within the 50% margin of clinically observed data. Nevertheless, an additional plasma clearance term could further improve the simulation outcome. The use of an additional clearance term is currently a limitation for predictions of de novo TPs because the additional clearance pathway requires fitting and, thus, observed data. It was demonstrated for three case studies that the allometric scaling of the additional clearance improved the predictions, but the physiological clearance pathways might not scale with body weight.

This work highlights the need for a more mechanistic understanding of the other elimination pathways beyond the degradation in vascular endothelial cells, which are generally neglected by PBPK models. One factor is APCs, which are highly pinocytotically active [32] and contain FcRn. Two studies in mice demonstrated the importance of APCs in IgG homeostasis, in which endothelial cells and APCs contributed almost equally to the degradation of IgG [33,34]. PBPK modeling demonstrated that APCs in lymph nodes, draining the SC site, explained the lower bioavailability due to the temporary saturation of the FcRn receptor after SC administration and consequently a higher catabolism rate [2]. The uptake, the binding to FcRn, and the subsequent degradation of TPs by APCs could therefore contribute to the overall clearance. Additionally, TPs could interact with Fcγ receptors that are expressed on the cells of the mononuclear phagocyte system. It was demonstrated that the polymorphism of Fcγ receptors had an impact on the therapeutic response to rituximab [35]. Fcγ receptors are typically involved in the receptor-mediated uptake of antigen complexes, which requires binding to a soluble target. Fcγ-R1 can also take up monomeric IgG [36].

Another important factor for TP clearance is pinocytosis into cells other than endothelial cells and APCs. Typically, every cell can take up surrounding fluid that contains molecules such as albumin, IgG, or exogenously administered TPs. Under physiological conditions, the highest exposure of TPs occurs in the blood vessels; thus, higher uptake is expected in endothelial cells surrounding the blood vessels and in APCs located in the systemic circulation. However, TPs can distribute via paracellular and transcellular pathways through the endothelial cell layer and be exposed to other cell types within the tissue. The paracellular pathway strongly depends on the number and sizes of the pores that connect the vascular space and the interstitial fluid. The liver and spleen have sinusoidal vessels with large gaps that lead to a rapid, well-stirred equilibrium between the vascular and interstitial concentrations. It was demonstrated for peptides that hepatocytes play a crucial role in degradation [37]. Furthermore, hepatocytes also contain FcRn to recycle endogenous IgG, which is not incorporated into PBPK models for TPs with an Fc domain or albumin. Despite the low concentration of TPs in the interstitial fluid of other organs and tissue types, such as the heart, muscle, and adipose tissue, cells such as myocytes and adipocytes can take up TPs by pinocytosis. As these cell types do not contain FcRn, TPs would be degraded in the lysosome and therefore would contribute to the total clearance, although this is expected to be minor compared with the vascular endothelium.

Mechanistic mathematical models do not account for these pathways because the mechanisms are still not fully understood and experimental data are lacking. The current PBPK models fit parameters such as the uptake rate, the catabolic clearance, or the recycling rate and the fraction recycled for the vascular endothelium [10,38]. Adding more parameters with uncertainty, such as pinocytosis into different cell types (e.g., hepatocytes, heart cells, or adipocytes), might lead to identifiability issues regarding single parameters, as, often, only plasma concentration is available. More scientific evidence is required to build more mechanistic models.

Understanding the mechanisms of the additional clearance pathways and their mechanistic incorporation into PBPK models could lead to better translation from preclinical species to humans to estimate the first-in-human dose and better prediction between populations. However, the uptake rate depends not only on the cell type but also on the characteristics of the TPs, such as the charge, hydrophobicity, and glycosylation, which are typically not available to parameterize the model sufficiently [38]. The cell membrane is negatively charged because of the phospholipid heads facing the outside and the glycocalyx layer on the outside of the cell. Positively charged molecules have a higher likelihood of approaching the cell closely enough to distribute inside via pinocytosis. However, the relationship between the charge of a protein and its uptake rate or clearance is not quantified and thus relies on fitting in PBPK models [39]. Hydrophobicity characteristics can also explain the different interactions with cells and constituents such as the extracellular matrix in the interstitial space. Glycosylated proteins can distribute into cells by receptor-mediated endocytosis—for instance, through the asialoglycoprotein receptor (ASGPR) [40]—depending on their glycosylation structures. The degradation after the receptor-mediated uptake of a glycosylated protein leads to additional clearance. More descriptive correlations between the charge or hydrophobicity and the pinocytosis rate are required to enable better predictions of novel antibodies.

The current study used only simulations in the monkey, and, even for the monkey, not all case studies had experimental data. Lower animal species such as humanized FcRn mice (Tg 32) and the development of a mouse PBPK model could enhance our understanding of TPs in humans. Humanized FcRn mice express human FcRn instead of mouse FcRn, providing a better prediction of the pharmacokinetics in humans. These Tg 32 mouse studies may offer more relevant data on drug clearance and distribution and help to assess mechanisms or elimination pathways not related to FcRn binding. Developing mouse PBPK models with these data and translating the model from mice to humans, or using them as an intermediate step from mice to monkeys with the allometric scaling of the additional systemic clearance, could confirm our mechanistic insights and increase the confidence in translating preclinical findings to human clinical settings [41,42,43].

The limitations of this work include the fact that most case studies focused on mAbs. Data on ADCs and Bi-TCEs were limited and there was no information for peptides or smaller TPs. However, it is reasonable to translate information gained from mAbs to ADCs and Bi-TCEs due to their similar physiological compositions and characteristics. Since the data came from a cross-industry collaboration, they demonstrate the current needs of the pharmaceutical industry.

## 5. Conclusions

Simulations in the monkey can be performed by using minimal data input, including the molecular weight and the FcRn binding affinity, which can be measured in human samples and scaled for simulations in the monkey, as well as the additional systemic clearance of IgG. Additional clearance is required for simulations of mAbs, ADCs, and Bi-TCEs in humans, which requires fitting to observed clinical data. We propose mechanisms to investigate the additional clearance pathways experimentally and include these in mechanistic mathematical models to achieve better translation and predictions.

## Figures and Tables

**Figure 1 pharmaceutics-17-00560-f001:**
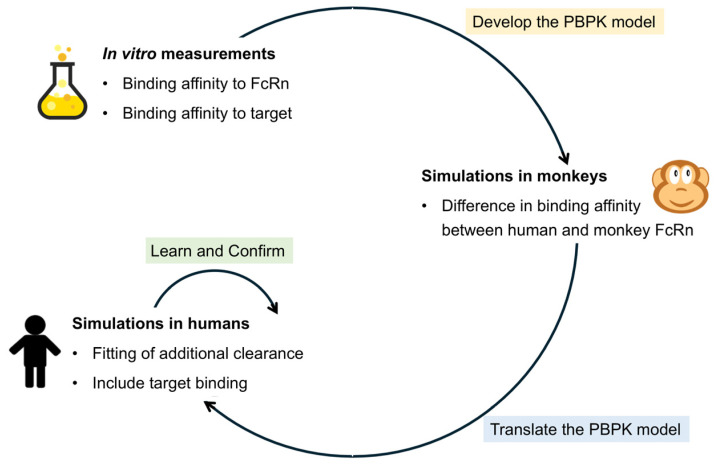
General workflow used in this study.

**Figure 2 pharmaceutics-17-00560-f002:**
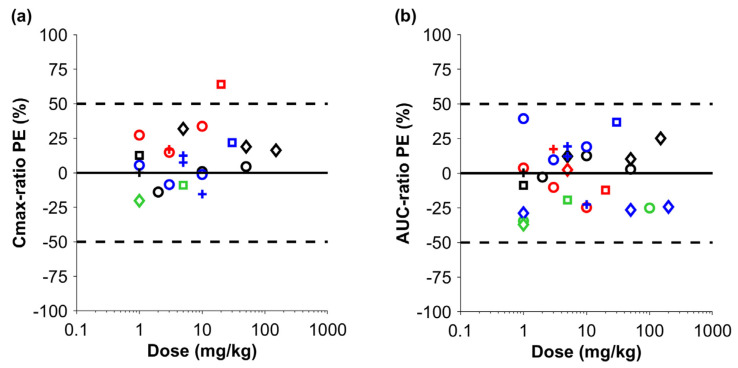
Prediction error (PE) for C_max_ (**a**) and AUC (**b**) in cynomolgus monkeys for biologic drugs. The solid black line and the dashed black lines show no *PE* and +/−50% PE, respectively. The dots represent the *PE* for each simulated scenario (black circles = [18], red circles = [19], blue circles = [20], green circles = [21], black diamonds = [22], red diamonds = [23], blue diamonds = [24], green diamonds = [25], black squares = [26], red squares = [27], blue squares = [28], green squares = [29], black plus sign = [30], red plus sign = [31], blue plus sign = case study). One drug could have multiple doses, as indicated in Appendix A.

**Figure 3 pharmaceutics-17-00560-f003:**
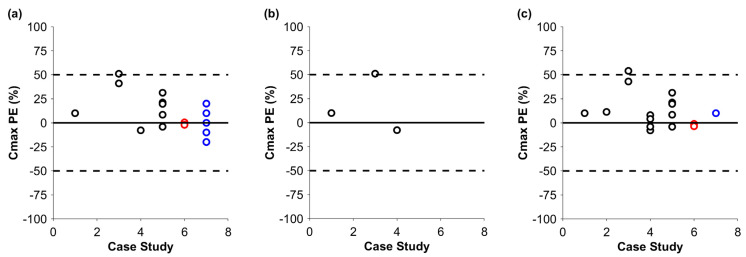
Prediction error (PE) for C_max_ in humans for biologic drugs without additional systemic clearance (**a**), with allometrically scaled additional systemic clearance (**b**), and with optimized additional systemic clearance (**c**). The solid black line and the dashed black lines show no *PE* and +/−50% PE, respectively. Black, red, and blue markers stand for the PEs of mAbs, ADCs, and Bi-TCEs. Simulations for the ADC and Bi-TCE were published previously [4,17].

**Figure 4 pharmaceutics-17-00560-f004:**
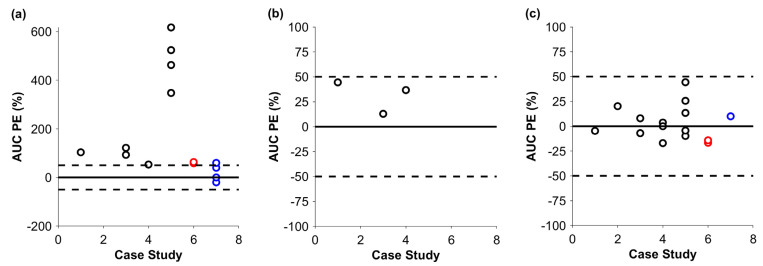
Prediction error (PE) for the AUC in humans for biologic drugs without additional systemic clearance (**a**), with allometrically scaled additional systemic clearance (**b**), and with optimized additional systemic clearance (**c**). The solid black line and the dashed black lines show no *PE* and +/−50% PE, respectively. Black, red, and blue markers stand for the PEs of mAbs, ADCs, and Bi-TCEs. Simulations for the ADC and Bi-TCE were published previously [4,17].

**Table 1 pharmaceutics-17-00560-t001:** Overview of the seven case studies. Simulations in the cynomolgus monkey were only performed when preclinical data were available (indicated by the green arrow). Simulations in humans were run without the additional systemic clearance (CL_add_), with allometrically scaled CL_add_ if simulations were performed in the monkey, and with an optimized CL_add_.

Performed Simulations	Case 1:mAb	Case 2:mAb	Case 3:mAb	Case 4:mAb	Case 5:mAb	Case 6:ADC	Case 7:Bi-TCE
Monkey	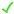	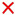	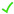	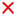	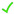	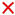	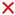
Human without CL_add_	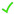	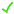	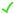	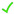	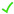	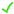	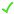
Human with allometrically scaled CL_add_	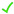	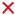	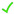	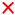	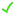	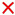	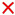
Human with optimized CL_add_	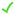	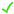	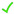	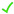	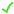	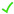	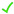

ADC = antibody–drug conjugate, Bi-TCE = bispecific T-cell engager, CL_add_ = additional systemic clearance, mAb = monoclonal antibody. Green arrow = data were available, red cross = data were not available.

**Table 2 pharmaceutics-17-00560-t002:** Input parameters based on the molecule type. Green arrows indicate that the input is required.

Parameter	mAb	ADC	Bi-TCE
Molecular weight (g/mol)	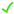	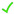	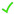
KD to FcRn (µM)	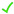	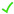	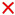
Additional systemic clearance (L/h)	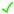	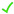	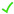
Renal filtration clearance	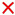	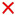	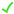
Deconjugation	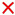	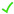	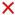

ADC = antibody–drug conjugate, Bi-TCE = bispecific T-cell engager, KD = binding affinity, mAb = monoclonal antibody. Green arrow = data were available, red cross = data were not available.

## Data Availability

The data presented in this study are available on request from the corresponding author due to commercial restrictions.

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
