# Peer review of "Physiologically Based Pharmacokinetic Modeling of Biologic Case Studies in Monkeys and Humans Reveals the Necessity of an Additional Clearance Term"

_pharmaceutics, 2025, doi:10.3390/pharmaceutics17050560_

Round 1

Reviewer 1 Report

Comments and Suggestions for Authors

This is an overall well written manuscript addressing the topic of clinical PK prediction of biologic drugs (mainly monoclonal antibodies, but also bi-specific T-cell engager and antibody-drug conjugate included) based on preclinical PK data in the monkey and physiologically-based pharmacokinetic modeling in Simcyp. The authors find that purely mechanistic (default PBPK-based) human PK prediction is currently insufficient, as optimization of human clearance based on clinical PK data has been required in case studies to obtain human PK predictions of acceptable accuracy. This involved definition of an additional clearance term.

The following comments may help to further enhance the understanding of the manuscript to readers and reproducibility:

  1. Methods: «additional 14 studies were found and used in the analysis». Suggest to combine information on these studies with Table 1.
  2. Table 1: in addition to comment 1, please make table more clear and readable without reading through the manuscript. For example what does «Data in monkey» mean? ( «PK data in monkeys available with developed PBPK model in monkeys»? ) The subsequent lines appear to connect to the «four different scenarios (line 139)», while only three scenarios are listed in the table. TMDD is mentioned in the legend of the table, but not used in the table.
  3. Methods, p.3: It would be helpful to summarize for the reader all input parameters and selected physiological values to be enterd in a table
  4. Line 121: each company… developed their own model. Was really the same (before described) initial implementation approach chosen in all companies (for monkeys? Humans?), or was the implementation harmonized for this project?
  5. Line 128: «… and to refine the model if required»: would it be possible to state exactly which components were considered to be refined?
  6. Line 138: «minimal input data»: please specify (could be explained in possible input parameter table, suggested under point 3).
  7. Figure 1: Workflow of the study. It looks like additionial in vitro measurements were performed for the presented analysis, but this appears to be not mentioned in the results section. A further extension of the legend, explaining how to read the illustration, would be helpful.
  8. Line 152: «simulations were performed… with sparse data» appears to describe methods. What is meant with «with sparse data»?
  9. Figure 2: Legend misses explanation of dots, PE a definition (probably =predicted/observed -1 x 100%?). Given the number of dots (>7) are there multiple predictions per compound? Could these «repeated predictions» be indicated on the figure?
  10. Figure 3: To which of the four scaling scenarios of human clearance do these predictions correspond to? It would help for communication if all scenarios would be illustrated as multi-panel figure. Legend also misses explanation of dots, given the number of dots (>7) there also appear to be multiple predictions per compound? Could these «repeated predictions» be indicated on the figure?

Author Response

Methods: «additional 14 studies were found and used in the analysis». Suggest to combine information on these studies with Table 1.

Thank you for your suggestion. The 14 additional studies were used to evaluate the predictions of humanized IgG in the monkey only. It would mean to only tick the first row (simulations in the monkey performed) but none of the other three rows (simulations in the humans without CLadd, with allometrically scaled CLadd, and with optimized CLadd) in table 1. We feel that the table would grow from 7 to 21 columns (or rows because we would turn the table) but there would not be more information, and it could even become more difficult to visualize the information for the different case studies. Table S2 gives an overview about the 14 studies with the molecule type, the doses used, and the reference. Additionally, we changed the sentence in the methods to make clear that the studies are for the monkey only.

Line 90: “Additional 14 studies for the pharmacokinetics of humanized IgG in the monkey were found in the literature and used in the analysis (Table S2).”

Table 1: in addition to comment 1, please make table more clear and readable without reading through the manuscript. For example what does «Data in monkey» mean? ( «PK data in monkeys available with developed PBPK model in monkeys»? ) The subsequent lines appear to connect to the «four different scenarios (line 139)», while only three scenarios are listed in the table. TMDD is mentioned in the legend of the table, but not used in the table.

Thank you for your comment. We have revised Table 1. The first column says now which simulations were performed. There are four possible simulations: In the monkey based on the availability of preclinical data, human simulations without the additional systemic clearance that was performed for all seven cases, human simulations with an allometrically scaled clearance that could only be conducted when simulations were performed in the monkey, and simulations in the human with an optimized additional systemic clearance that was performed for all cases. The table legend was updated to reflect the changes.

Additionally, the sentence in line 254 was changed because only three scenarios were simulated in humans and were described in this paragraph. Simulations, considering TMDD were not required, because the doses were high enough to saturate the target.

Methods, p.3: It would be helpful to summarize for the reader all input parameters and selected physiological values to be entered in a table

Thank you for your suggestion. Table 2 was added to show the input parameters, depending on the molecule type. Physiological parameters were not altered in the simulations.

Line 121: each company… developed their own model. Was really the same (before described) initial implementation approach chosen in all companies (for monkeys? Humans?), or was the implementation harmonized for this project?

Thank you for your comment. The compound models were developed by each company. We harmonized the different required scenarios in humans. Simulations should have been conducted without additional systemic clearance, with allometrically scaled additional systemic clearance (if monkey data were available), and with an optimized additional systemic clearance.

Line 128: «… and to refine the model if required»: would it be possible to state exactly which components were considered to be refined?

Thank you for your comment. We agree that this sentence is misleading. The statement was more general for the workflow. The paragraph was revised to distinguish between the general workflow and what was done in the study.

The only parameter that was optimized in this study was the additional systemic clearance. However, in the general workflow it is possible to also optimize other parameters such as the endocytosis uptake rate. The uptake rate can be drug-specific, based on for instance electrostatic properties of the studied protein. The plasma membranes are negatively charged because of the phospholipids facing the outside. A positively charged protein has a higher likelihood of getting close enough to the cell membrane to distribute into the endosome by fluid-phase endocytosis whereas negatively charged proteins might have a lower likelihood. This can be expressed in a PBPK model by using different uptake rates. A protein that distributes into the cell will further distribute into the lysosome and be degraded by proteases if FcRn binding does not occur. A higher uptake rate can result in a higher catabolic clearance. However, there is no quantitative relationship between the charge of a protein and the uptake rate. We decided to look only into the additional systemic clearance that can include the process of uptake into non-FcRn expression cells with consequent degradation.

Line 154: “If predictions do not match the observed data, the learn and confirm approach can be used to build a more robust mechanistic model by doing additional in vitro measurements. Alternatively, processes that are not well understood such as the additional systemic clearance can be optimized. In this study, additional in vitro data were not obtained but the additional clearance was investigated in the human simulations”

Additionally, we updated figure 1 to show only the workflow of the study and no longer the general workflow.

Line 138: «minimal input data»: please specify (could be explained in possible input parameter table, suggested under point 3).

Thank you for your suggestion. The minimal input data are the molecular weight, the binding affinity to FcRn (measured for human FcRn and scaled to monkey FcRn), and the additional systemic clearance of IgG (meaning 20% of the systemic clearance). We extended the sentence to make clear what we mean with minimal input parameters. And we introduced Table 2 to have a summary of required parameters.

Line 168: “Parameterization of the biologics drug in the monkey model was done with minimal input data, including the molecular weight, the binding affinity to FcRn, and with the additional systemic clearance similar of IgG”

Figure 1: Workflow of the study. It looks like additional in vitro measurements were performed for the presented analysis, but this appears to be not mentioned in the results section. A further extension of the legend, explaining how to read the illustration, would be helpful.

Thank you for your comment. This was meant to a general workflow, but no additional experimental data were gathered in this study. We extended the explanation for the workflow and made clear that no additional experiments were performed. Additionally, the figure was changed to reflect the workflow used in this study.

Line 154: “If predictions do not match the observed data, the learn and confirm approach can be used to build a more robust mechanistic model by doing additional in vitro measurements. Alternatively, processes that are not well understood such as the additional systemic clearance can be optimized. In this study, additional in vitro data were not obtained but the additional clearance was investigated in the human simulations.”

Line 152: «simulations were performed… with sparse data» appears to describe methods. What is meant with «with sparse data»?

Thank you for your comment. We have extended the explanation what we mean with sparse data. It included the molecular weight, the binding affinity to human FcRn scaled to monkey FcRn, and the additional systemic clearance of IgG. Scaling of the binding to monkey FcRn was done according to the data of Abdiche et al (Abdiche et al., 2015; mAbs 7(2): 331-343). They reported that human IgG binds to human FcRn with a KD of 737 nM at pH 5.8. The KD of human IgG to monkey FcRn was 332 nM and therefore a factor of 0.45 applies. KD to human FcRn was therefore multiplied with 0.45 as explained in line 152.

Line 198: “Firstly, simulations were performed in cynomolgus monkeys with sparse data, including the molecular weight, the binding affinity to human FcRn scaled to the monkey and the additional systemic clearance of IgG (Figure 2).”

Figure 2: Legend misses explanation of dots, PE a definition (probably =predicted/observed -1 x 100%?). Given the number of dots (>7) are there multiple predictions per compound? Could these «repeated predictions» be indicated on the figure?

Thank you for your comment. The figure was changed so that each drug is represented by a different symbol/colour. The different doses are explained in Table S2. The explanation for the dots was added to the figure legend. The definition of the prediction error was introduced into the methods (line 187).

Figure 3: To which of the four scaling scenarios of human clearance do these predictions correspond to? It would help for communication if all scenarios would be illustrated as muli-panel figure. Legend also misses explanation of dots, given the number of dots (>7) there also appear to be multiple predictions per compound? Could these «repeated predictions» be indicated on the figure?

Thank you for your comment. Yes, each case could have more simulations caused by the number of doses or similar molecules. Table S1 in the supplementary material was updated to inform how many simulations were conducted for each case. Figure 3 was updated and split into two different figures (Figure 3 for Cmax and Figure 4 for the AUC). Each figure shows three different subplots: one for the scenario without an additional systemic clearance, one for the allometrically scaled clearance, and one for the optimized clearance. Additionally, the figure legend was updated to include the information about the dots.

Reviewer 2 Report

Comments and Suggestions for Authors

1) The Authors performed an article entitled "Physiologically based pharmacokinetic modelling of biologic case studies in monkeys and humans reveals the necessity of an additional clearance term". The objectives of the study were to establish a workflow to develop a PBPK model for biologic drugs; to analyse if PBPK models for cynomolgus monkeys can be developed with sparse model parameterization; and to investigate the need for an additional systemic clearance in humans. The publication has therapeutic interest and contains 3 figures, 1 table, 29 bibliographic references, and supplementary material.

2) Line 78: “Error! Reference source not found”?

3) Lines 136/137: Simulations were first conducted in a typical cynomolgus monkey, weighting 4 kg. Why 4 kg? How do results vary in male and female animals? Please justify your answers.

4) Line 175: “Error! Reference source not found”?

Author Response

Line 78: “Error! Reference source not found”?

Thank you for your comment. It appears that the cross-reference to Table 1 caused the error in line 78 and in line 175. The cross-referencing was fixed.

Lines 136/137: Simulations were first conducted in a typical cynomolgus monkey, weighting 4 kg. Why 4 kg? How do results vary in male and female animals? Please justify your answers.

Thank you for your comment. We have run the simulation in a single male monkey representative and changed the text to make it clear to the reader.

Line 167: “Simulations were first conducted in a typical male cynomolgus monkey, weighting 4 kg.”

Most observed data were obtained in male cynomolgus monkeys. Two of the paper investigated differences in male and female cynomolgus monkeys and did not find a sex-dependent pharmacokinetics of IgG-like antibodies in the animals (Deng et al., 2010; Köck et al., 2016). The body weight of 4 kg is included in the Simcyp Animal Simulator and is the result of an analysis of the typical body weight of a cynomolgus monkey in pharmaceutical research.

Line 175: “Error! Reference source not found”?

We are sorry for the inconvenience. As said above, the cross-reference to Table 1 must have caused this error and this was fixed.

Reviewer 3 Report

Comments and Suggestions for Authors

In the present manuscript, the authors proposed a workflow for PBPK modeling of various biologics and incorporation of additional clearance term. Overall, the manuscript is well written but the authors should consider following comments to address critical gaps in the manuscript for the benefit of the reader.

  • Introduction, line 44: The PBPK approach can support the drug design from a physiological perspective. This sentence is not very clear. Please elaborate on specific applications.
  • Introduction, line 58: the authors mentioned that binding is often measured for human FcRn but not in preclinical species. Please elaborate probable reasons for the same. If this data is not available, how does the preclinical modeling is executed
  • Materials & methods, line 77: please clarify if all the 7 case studies are intravenous only or if there are other routes of administration
  • Materials & methods, line 109: the FcRn expression (per gram) was assumed in monkeys to be same as that of humans. This is big assumption and please discuss the results in the context of this assumption. Can the authors carry certain sensitivity analysis around this to evaluate impact on predictability.
  • Materials & methods, line 123: was the FcRn binding to pH 7.0 was considered in the model as it may account for interstitial binding in the cell?
  • Materials & methods, line 141: one of the approach taken by authors was allometry. Was single species was used for allometry? Or multiple species. Please elaborate this aspect
  • Materials & methods, line 158: the authors calculated prediction error to validate simulations. However the acceptability was based on 1.5 folds margin. Thus this is not exactly prediction error but folds error
  • Materials & methods: Did the authors consider TMDD or any other disposition process in the model? Please clearly segregate modeling section to include inputs, biological processes consideration and elimination pathways consideration. This segregation can provide clarity to the reader
  • Materials & methods: Can authors include a table providing model input parameters, this helps reader to reproduce these models
  • Results, Figure 2: there is slight trend of over prediction for Cmax as the dose increases. Was there any finding about this? Also can the authors add corresponding biological moiety type (e.g. ADC, mAB etc) to understand more about this pattern?
  • Results: Can the authors clarify what exactly is the additional clearance term and its physiological relevance? Unless and until this is explicitly explained in the manuscript, it may be little difficult to justify. How does the authors suggest consideration of this additional term for different biological moieties (e.g. ADC, mAB etc)
  • Discussion: in the present case, majority of case studies had IV as administration and one case study has SC. Are the observations regarding additional clearance term is similar for both types? Can authors provide some discussion around this?
  • Discussion: the discussion also can be structured to indicate modeling or workflow suggestions for specific biological moiety (e.g. ADC, mAB etc).

Author Response

Introduction, line 44: The PBPK approach can support the drug design from a physiological perspective. This sentence is not very clear. Please elaborate on specific applications.

Thank you for your comment. What we meant is that PBPK modelling cannot only be used late in drug development to predict DDIs or the pharmacokinetics in populations with limited clinical data (i.e. organ impairment, children) but also earlier in drug development. For instance, if the molecule is engineered and target binding affinity is altered, then the measured target binding affinity can be entered into a PBPK model and the alterations of the drug characteristics can be evaluated under physiological conditions and thus, the PBPK approach can support the drug design from a physiological perspective. We have added the example to the sentence:

Line 45: “The in vitro measured drug characteristics such as the target binding affinity can be entered into a PBPK model to predict the result of any engineered modifications of the drug under physiological conditions and thus, the PBPK approach can support the drug design from a physiological perspective”.

Introduction, line 58: the authors mentioned that binding is often measured for human FcRn but not in preclinical species. Please elaborate probable reasons for the same. If this data is not available, how does the preclinical modeling is executed

Thank you for your question. We at Certara have asked pharmaceutical companies about the data that are measured for an antibody and the result was that the binding affinity is often determined for human FcRn only. We did not ask for the specific reasons, but potentially it is related to capacity and costs. Abdiche et al. published the binding affinity of IgGs from different species to FcRn from different species (Abdiche et al., 2015; mAbs 7(2): 331-343). They reported that human IgG binds to human FcRn with a KD of 737 nM at pH 5.8. The KD of human IgG to monkey FcRn was 332 nM and was therefore a factor 0.45 lower compared with the binding to human FcRn. If we assume that it holds true for all IgG-based antibodies, we can use the factor of 0.45 to convert the measured KD to human FcRn for each antibody.

We have added the reason for measuring only the binding affinity in humans to the sentence in the introduction.

Line 62: “During the development of novel biologic drugs, the binding is often only measured for human FcRn because of capacity and cost reasons and appropriate data are lacking in animals.”

The conversion of measured KD values to human FcRn to monkey FcRn is described in the methods section:

Line 151: “The species-dependent difference of IgG binding [8] was assumed to hold true in all cases, meaning measured human KD values were multiplied with 0.45 to describe the binding to monkey FcRn”

Materials & methods, line 77: please clarify if all the 7 case studies are intravenous only or if there are other routes of administration

Thank you for your comment. The mAbs and the ADC were administered IV and the Bi-TCE were administered SC. A sentence was added to the methods part to clarify the route of administration.

Line 84: “The mAbs and the ADC were administered by intravenous infusion (IV) and the Bi-TCE were administered SC”.

Additionally, Table S1 in the supplementary material contains the route of administration.

Materials & methods, line 109: the FcRn expression (per gram) was assumed in monkeys to be same as that of humans. This is big assumption and please discuss the results in the context of this assumption.

Can the authors carry certain sensitivity analysis around this to evaluate impact on predictability.

Thank you for your comment. In the absence of data, we assumed that the tissue concentration of FcRn per g tissue is the same between monkey and humans, meaning the difference in tissue weight is accounted for. In previous PBPK models for the monkey, the fitted human FcRn concentration was assumed to hold true for the monkey (Shah & betts, 2012; J Pharmacokineti Pharmacodyn 39: 67-86; Glasman et al., 2015; J Pharmacokinet Pharmacodyn 42(5): 527-540). We are using the same approach but not with a fitted FcRn value. A sensitivity analysis will show that FcRn abundance is a highly sensitive parameter, similar to other parameters of the FcRn salvage pathway such as the pinocytosis uptake rate (Kup). Parameters of the FcRn salvage pathway such as the recycling rate and the fraction recycled have not been measured and thus, require fitting against IgG. We wanted to reduce the number of parameters used for fitting to decrease the uncertainty and have thus, used the measured FcRn values for the monkey, but the other parameters were determined using those FcRn values. Li et al. demonstrated that multiple parameter combinations led to a good fit between predicted and observed plasma concentrations (Li et al., 2014; AAPS 16(5): 1097-1109). The use of a different FcRn abundance will likely result in a difference in the other parameters such as the fraction recycled. In the performance evaluation, we orientate towards IgG half-life, which was reported to be 8.3 days in rhesus monkeys (Challacombe & Russell 1979; Immunology 36(2): 331) and was predicted with 8.1 days in the monkey PBPK model. Additionally, independent observed data of humanized IgGs are well predicted as shown in the figure.

Figure: The black solid line shows the prediction of humanized IgG in cynomolgus monkeys. The red markers are from the observed studies (see Table S2 in the supplementary material for reference).

We argue that the use of measured human FcRn is indeed an assumption, but the predictions appear to be reasonable accurate. Of course, measured data of FcRn abundance in the monkey would be preferable.

Materials & methods, line 123: was the FcRn binding to pH 7.0 was considered in the model as it may account for interstitial binding in the cell?

Thank you for your comment. IgG typically does not bind to FcRn at neutral pH (Vaughn et al., 1998; Structure 6(1): 63-73) and thus, binding at pH 7.0 is not considered in the model. Under physiological conditions, the IgG-FcRn complex distributes either back to the blood vessels or to the interstitial space, in which IgG is released, leading to the long half-life of IgG. The absence of binding at neutral pH is essential for the release. Differences are only expected for engineered molecules such as FcRn antagonists, but no FcRn antagonist was modelled in this study and thus, the model structure can be simplified.

Materials & methods, line 141: one of the approach taken by authors was allometry. Was single species was used for allometry? Or multiple species. Please elaborate this aspect

Thank you for your comment. Allometric scaling of the additional systemic clearance was only done if preclinical data were available in the monkey and was therefore only done from a single species. We argue in the discussion that the inclusion of other species such as rodents could lead to better predictions, but the main point of improvement is a better understanding of the clearance pathways that we described with the additional systemic clearance term.

Materials & methods, line 158: the authors calculated prediction error to validate simulations. However, the acceptability was based on 1.5 folds margin. Thus this is not exactly prediction error but folds error

Thank you for your comment. We have corrected it to be within -50% to +50%, which is indeed not equivalent to the 1.5-fold interval.

Line 189: “A PE between the -50% to +50% margin was considered acceptable.”

The correction was made throughout the manuscript.

Materials & methods: Did the authors consider TMDD or any other disposition process in the model? Please clearly segregate modeling section to include inputs, biological processes consideration and elimination pathways consideration. This segregation can provide clarity to the reader

Thank you for your comment. The paragraph in the methods was rephrased to make the used distribution and clearance processes clearer.

Line 112: “Distribution processes include the two-pore hypothesis to calculate the paracellular distribution between the vascular and the interstitial space [5]. Parameters for the two-pore hypothesis, including the determination of pore sizes and number of pores, were described previously [9]. The transcellular distribution pathway consists of a linear rate, representing macropinocytosis into the endothelial cell layer, binding to FcRn at pH 6.0, and recycling of the bound complex to the vascular space or transcytosis to the interstitial space. The pinocytosis uptake rate was assumed to be the same for all tissues and was set to 0.0298 1/h based on published data with horse radish peroxidase, a common pinocytosis marker [11]. Tissue-specific FcRn abundance for humans was taken from the literature [12] and the same values per gram tissue were assumed to hold true in the monkey in the absence of relevant data. Tissue-specific FcRn abundance in the monkey was used to calculate the total body FcRn concentration, which was used in the monkey simulations. Recycling rate was fitted to match the plasma concentration and half-life of exogenous IgG in monkeys [13] and humans [14].

Three unspecific elimination pathways were considered. Firstly, catabolism in endothelial cells of the unbound IgG-like antibody [10]. The contribution of each tissue to the catabolism was taken from the literature [15, 16]. Secondly, an additional plasma clearance which represents processes that are not mechanistically accounted for such as the catabolism in antigen-presenting cells (APCs) or pinocytosis into non-FcRn expressing cells with subsequent catabolism. The additional systemic clearance of IgG was set to 20% of the total IgG clearance, based on previous findings [10]. Monkey simulations used the default value for the additional systemic clearance of IgG to investigate the minimal parameter requirement, which was set to 20% of total IgG clearance. Thirdly, renal filtration, which was only used for the Bi-TCE model [17].”

Monkey was said to be a non-cross-reactive species for all investigated antibodies and thus, TMDD was not considered in the monkey simulations. A sentence was added to the workflow of the monkey simulations.

Line 171: “Monkey was a non-cross-reactive species for all investigated antibodies and thus, TMDD was not considered in the simulations.”

To fit the additional systemic clearance in the human simulations, other clearance processes such as catabolism in endothelial cells or renal filtration need to be accounted for. Additionally, there should not be any contribution from the clearance, mediated by target binding. Thus, doses in the human simulation were high enough to saturate the target. TMDD was not the clearance-determining process.

Line 177: “To optimize the additional systemic clearance, the dose had to be high enough to saturate the target and thus, TMDD was not the clearance-determining process.”

We also added a sentence to inclusion criteria to say that at least one dose had to be high enough to saturate the target and allow for the optimization of the additional systemic clearance.

Line 80: “At least one dose had to be high enough to saturate the target to allow the optimization of the additional systemic clearance.”

Materials & methods: Can authors include a table providing model input parameters, this helps reader to reproduce these models

Thank you for your comment. Table 2 was added to show the required input parameters for the PBPK model.

Results, Figure 2: there is slight trend of over prediction for Cmax as the dose increases. Was there any finding about this? Also can the authors add corresponding biological moiety type (e.g. ADC, mAB etc) to understand more about this pattern?

Thank you for your comment. Figure 2 show the results for the monkey, which were only available for mAbs. We have modified the figure according to a comment from reviewer 1 to show different doses of the same mAb to make the figure clearer.

All mAbs in Figure 2 were injected IV and thus, the plasma volume is a key physiological parameter to determine the Cmax. Plasma volume depends on the body size and thus, a difference in body weight between model simulations and the observed data can explain a difference in predicted vs observed Cmax. Another reason is the time point of the sampling in a preclinical study, because measurements would need to be taken immediately after the end of the infusion because otherwise the true Cmax is not measured.

Results: Can the authors clarify what exactly is the additional clearance term and its physiological relevance? Unless and until this is explicitly explained in the manuscript, it may be little difficult to justify. How does the authors suggest consideration of this additional term for different biological moieties (e.g. ADC, mAB etc)

Thank you for your comment. We have decided to explain the additional systemic clearance in the discussion rather than the result section as we feel that it is not the result of our work but work that is required in the future. Additionally, we added an interpretation in the method section, when the additional systemic clearance is introduced.

Line 136: “Secondly, an additional plasma clearance which represents processes that are not mechanistically accounted for such as the catabolism in antigen-presenting cells (APCs) or pinocytosis into non-FcRn expressing cells with subsequent catabolism.”

The additional systemic clearance describes linear clearance pathways that are not mechanistically accounted for in the PBPK model because of a lack of experimental data and understanding of the physiological processes. More concretely, the clearance pathways are 1) catabolism in antigen-presenting cells (APCs), and 2) pinocytosis and degradation in other cell types than APCs and endothelial cells.

APCs contribute to the IgG homeostasis as demonstrated by preclinical data in mice (Richter et al., 2018; mAbs 10(5): 803-813; Challa et al., 2019; mAbs 11(5): 848-860). The unknown parameters are the FcRn distribution between APCs and endothelial cells (currently only total FcRn per tissue are available for humans), and the question whether recycling parameters are similar between APC and endothelial cells. The parameters could be fitted in the model, but it would increase the number of fitted parameters and would lead to identifiability issues.

Pinocytosis is a process that can occur in all cell types. Endothelial cells, surrounding the blood capillaries, are highly exposed to the drug in the systematic circulation and thus, there is a higher likelihood of pinocytosis uptake into endothelial cells. However, biologic drugs distribute into the interstitial space through the two-pore process and therefore, other cell types facing the interstitial space can also uptake biologic drugs by pinocytosis. If the cells do not express FcRn, it would lead to catabolism. The missing parameter are the uptake rate, because it could depend on the cell type. Fitting the data towards plasma concentration would lead to identifiability issues and thus, measured data are required.

The additional clearance was required for all investigated biologic modalities. The described processes (pinocytosis uptake and consequent catabolism in APCs and in other cells than APCs and endothelial cells) occur independent of the molecule type. In all cases, clinically observed data are required for the model fit. To build a true bottom-up model, more experimental data are required as suggested in the discussion.

Discussion: in the present case, majority of case studies had IV as administration and one case study has SC. Are the observations regarding additional clearance term is similar for both types? Can authors provide some discussion around this?

Thank you for your question. The development of a PBPK model starts ideally from IV data, if available, so that the distribution and clearance processes can be evaluated without the absorption processes. Doses, that are high enough to saturate the target, are used to fit the additional systemic clearance after IV dosing. In a next step, the model is switched to subcutaneous administration, including the absorption and presystemic clearance processes, which are independent of the additional systemic clearance. Thus, the additional systemic clearance is independent of the route of administration. However, if only SC data are available like for Bi-TCE case example, then the additional systemic clearance can only be fitted towards plasma concentration after SC dosing.

The interesting aspect of this work is that all simulated modalities required an additional systemic clearance.

Discussion: the discussion also can be structured to indicate modeling or workflow suggestions for specific biological moiety (e.g. ADC, mAB etc).

Thank you for your comment. The workflow is generic and can be applied to all investigated molecule types. Furthermore, our study revealed the need for an additional systemic clearance term for all investigated modalities and thus, there is no specific suggestion for single molecule types. We made the conclusion clearer by mentioning the investigated biologic modalities.

Line 349: “Additional clearance is required for simulations of mAbs, ADCs, and Bi-TCE in humans, which requires fitting to observed clinical data.”

Round 2

Reviewer 3 Report

Comments and Suggestions for Authors

I would like to thank authors for answering all questions appropriately. I don't have further comments.